# Stepwise stroke recognition through clinical information, vital signs, and initial labs (CIVIL): Electronic health record-based observational cohort study

Sung Eun Lee[1,2☯], Mun Hee Choi[1☯], Hyo Jung Kang[2], Seong-Joon Lee[1], Jin Soo Lee[1], Yunhwan Lee[3], Ji Man Hong[1] *

1 Department of Neurology, Ajou University School of Medicine, Ajou University Medical Center, Suwon, Republic of Korea, 2 Department of Emergency Medicine, Ajou University School of Medicine, Ajou University Medical Center, Suwon, Republic of Korea, 3 Department of Preventive Medicine & Public Health, Ajou University School of Medicine, Suwon, Republic of Korea

☯ These authors contributed equally to this work.
* dacda@hanmail.net

## Abstract

### Background

Stroke recognition systems have been developed to reduce time delays, however, a comprehensive triaging score identifying stroke subtypes is needed to guide appropriate management. We aimed to develop a prehospital scoring system for rapid stroke recognition and identify stroke subtype simultaneously.

### Methods and findings

In prospective database of regional emergency and stroke center, Clinical Information, Vital signs, and Initial Labs (CIVIL) of 1,599 patients suspected of acute stroke was analyzed from an automatically-stored electronic health record. Final confirmation was performed with neuroimaging. Using multiple regression analyses, we determined independent predictors of tier 1 (true-stroke or not), tier 2 (hemorrhagic stroke or not), and tier 3 (emergent large vessel occlusion [ELVO] or not). The diagnostic performance of the stepwise CIVIL scoring system was investigated using internal validation. A new scoring system characterized by a stepwise clinical assessment has been developed in three tiers. **Tier 1:** Seven CIVIL-$AS^3A^2P$ items (total score from –7 to +6) were deduced for true stroke as $A$ge ($\geq$ 60 years); $S$troke risks without $S$eizure or psychiatric disease, extreme $S$ugar; "any $A$symmetry", "not $A$mbulating"; abnormal blood $P$ressure at a cut-off point $\geq$ 1 with diagnostic sensitivity of 82.1%, specificity of 56.4%. **Tier 2:** Four items for hemorrhagic stroke were identified as the CIVIL-$MAPS$ indicating $M$ental change, $A$ge below 60 years, high blood $P$ressure, no $S$troke risks with cut-point $\geq$ 2 (sensitivity 47.5%, specificity 85.4%). **Tier 3:** For ELVO diagnosis: we applied with CIVIL-$GFAST$ items ($G$aze, $F$ace, $A$rm, $S$peech) with cut-point $\geq$ 3 (sensitivity 66.5%, specificity 79.8%). The main limitation of this study is its retrospective nature and require a prospective validation of the CIVIL scoring system.

**Data Availability Statement:** All relevant data are within the paper and its Supporting Information files.

**Funding:** This study was supported by the Korea Centers for Disease Control and Prevention (KCDC, M2020-A0258-00012) and the Korean Stroke Society (KSS).

**Competing interests:** The authors have declared that no competing interests exist.

**Abbreviations:** CIVIL, Clinical Information, baseline Vital sign, and Initial Labs; CPSS, Cincinnati Prehospital Stroke Scale; CT, computed tomography; ELVO, emergent large vessel occlusion; EMR, Electronic Medical Record; EMS, emergency medical service; ER, emergency room; GFAST, gaze to face-arm-speech-time; LAPSS, Los Angeles Prehospital Stroke Screen; OR, odds ratio; ROC, receiver operating characteristic; ROSIER, Recognition Of Stroke In the Emergency Room.

## Conclusions

The CIVIL score is a comprehensive and versatile system that recognizes strokes and identifies the stroke subtype simultaneously.

## Introduction

Stroke remains with a high burden in societies, and improving the recognition of a stroke can help reduce this burden [1]. "Time-is-brain" is a crucial concept in ischemic stroke management [2,3], the importance of early recanalization has been further addressed after the success of endovascular recanalization therapies [4]. In the case of hemorrhagic stroke, early surgical interventions can be beneficial in selected patients [5,6]. Candidates for urgent interventions should be transported to an appropriately-equipped hospital, however, treatment can be delayed due to various reasons [7].

Stroke recognition systems have been developed to reduce time delay in community and hospital settings [8,9]. Various scales and scoring systems have been developed, but there is still no consensus on which scale perform better. [10,11]. Previous systems have issues related to false positives and false negatives, making it difficult for stroke specialists to handle a considerable number of patients [12,13]. In addition, most research has covered only one aspect of stroke, 'true stroke or not' or 'selection of emergent large vessel occlusion (ELVO),' and clinical parameters can be subjective even after thorough training. Finally, previously published systems have a greater focus on reducing pre-hospital delay [14,15]. Therefore, a comprehensive triaging system for considering next-step treatments should address to reduce the workload at stroke centers with acceptable sensitivity and specificity.

In this context, we have developed a new scoring system using Clinical Information, baseline Vital sign, and Initial Labs (CIVIL). Here, we aimed to evaluate the feasibility of the CIVIL system and compare with previous screening systems in suspicious acute stroke patients.

## Methods

### Study population

This electronic health record-based observational cohort study was performed in a tertiary referral hospital from January 2012 to December 2015. Care in the tertiary stroke center fulfilled the Brain Attack Coalition's standardized criteria, and the stroke unit had also obtained certification from the Korean Stroke Association [16]. The regional emergency medical center serves the southern part of Gyeonggi Province of South Korea with a population of approximately four million, and it's emergency room (ER) has approximately 89,000 patients annually [17]. Previously, we developed the stroke recognition system 'Cubic S model' (S1 Fig), which is based on the Electronic Medical Record (EMR) system for suspected stroke patients in the ER [18]. It is based on common signs and symptoms and contains three domains: time, body-part involvement, and symptomatic presentations. When suspicious stroke patients visit the ER, ER physicians check the presence of three domains: sudden onset, one-sided involvement of face/arm/leg, and 6 representative symptoms of stroke.

Each dataset was automatically stored in the database from a prospectively registered critical pathway system for rapid thrombolysis in suspicious stroke patient. The data has been used to improve the quality of registered data through monthly reports. Inclusion criteria for this study were (a) acute neurologic manifestations within 6 hours, (b) acute thrombolysis code

activating cases who meets all three domains of EMR-based Cubic S system, and (c) final confirmation with clinical and imaging findings by stroke neurologists. Exclusion criteria were: (a) incorrect activation of an acute thrombolysis code, (b) onset-to-door time > 6 hours, and (c) uncertain final diagnosis due to incomplete study. The study protocol was approved by the Institutional Review Board of Ajou University Hospital (AJIRB-MED-MDB-16-407). Informed consent was waived because of the study's retrospective nature.

## Processing after critical pathway activation

Initial assessment and activation of acute thrombolysis code was performed by ER physicians. Education of ER physicians and nurses was routinely performed every 6 to 12 months by stroke neurologists. Immediately after the activation of acute thrombolysis code, stroke neurologists assessed the patients. All patients except those with contraindications to contrast use underwent computed tomography (CT) scan with angiography. Simultaneously, neurologists meticulously investigated clinical information, baseline vital signs, initial laboratory findings, and stroke images. In cases where recanalization treatments were needed, intravenous recombinant tissue plasminogen activator and/or endovascular therapies were implemented according to critical pathway in our institute. Clinical information was recorded including age, sex, prior medical histories (hypertension, diabetes mellitus, previous stroke occurrence, seizure or syncope, and psychiatric history) from the patient, care-givers, or paramedics. Initial neurological manifestations were assessed with the Cubic S model. Baseline vital signs were comprised of blood pressure, pulse rate, and body temperature, and initial laboratory findings (glucose level and oxygen saturation) were also included.

## Confirmation of final diagnosis

The adjudication meeting comprised of stroke specialists was held weekly for final diagnoses of all patients according to the three tiers; as stroke mimic vs. true stroke, ischemic vs. hemorrhagic stroke, non-ELVO vs. ELVO. Final diagnosis was determined after review of ER chart, imaging and laboratory studies for differential diagnoses. True stroke was diagnosed when the neurologic exam was compatible with supportive imaging evidence of CT and/or magnetic resonance imaging including diffusion weighted image. Transient cerebral ischemic attack was classified into the true stroke and ischemic stroke group. Stroke mimics were designated when the clinical details were compatible with non-vascular etiologies. Initial CT angiography confirmed hemorrhagic stroke and large artery occlusion. ELVO was designated as occlusion of the internal carotid artery, M1 or M2 segment of the middle cerebral artery, or basilar artery [19,20].

## Development of new scoring system

We intended to develop a new scoring system that is characterized by a "stepwise clinical assessment system which enables rapid discrimination of patients suspected of acute stroke," potential variables were assessed including Clinical Information, Vital signs, and Initial Labs (CIVIL) used in ER and prehospital settings. There were 23 clinical findings, four vital signs, and two laboratory findings. We analyzed these variables by the three tiers; stroke mimic vs. true stroke, ischemic vs. hemorrhagic stroke, non-ELVO vs. ELVO. At the first tier (stroke mimic vs. true stroke), we assigned +1 to positive variables for true stroke (odds ratio [OR] > 1.0) and -1 to negative variables (OR <1.0). In the second tier, items suggestive of hemorrhagic stroke were derived (only positive scores). Finally, to discriminate ELVO from non-ELVO, we applied the gaze to face-arm-speech-time (GFAST) scoring system [21]. For evaluation of performance, the CIVIL system was compared with three previous recognition systems for acute

stroke: Cincinnati Prehospital Stroke Scale (CPSS) [14], Los Angeles Prehospital Stroke Screen (LAPSS) [15], and Recognition Of Stroke In the Emergency Room (ROSIER) system [8].

## Statistical analysis

Differences between the two groups at each steps were analyzed using $\chi^2$ or Student $t$-test for categorical and continuous variables, respectively. Significant variables from univariate analyses (p<0.05) were assessed with multivariate logistic regression models for deduction of scoring items (enter method). Associations were presented as odds ratios (OR) with corresponding 95% confidence interval (CI). Internal validation using receiver operating characteristic (ROC) curve analysis was performed to determine the optimal cut off point for each steps. Diagnostic performance including sensitivity, specificity, positive predictive value, negative predictive value, and Youden index were assessed for each cut-off point. We performed all analyses using SPSS 25.0 for Windows (SPSS Inc., Chicago, Ill).

# Results

## Patient assessment

The flow chart of study population is shown in S2 Fig. A total of 1,621 patients were screened by acute thrombolysis code activation. Sixty-two patients were excluded, and the remaining 1,559 suspected stroke patients were enrolled, of these, true stroke was confirmed in 1,153 (74.0%). Causes of stroke mimicking symptoms were metabolic disease (18.0%), drug intoxication (15.0%), peripheral neuropathy (14.3%), psychogenic disorder (14.3%), seizure (13.8%), infectious disease (7.4%), syncope (6.9%), and tumorous condition (3.4%). True stroke patients comprised of ischemic stroke (n = 894, 77.5%) and hemorrhagic stroke (n = 259, 22.5%), and the number of ischemic stroke patient requiring recanalization therapy was 291 (32.6%).

## Clinical information, baseline vital signs, and initial labs (CIVIL)

Tables 1 and 2 summarizes detailed findings according to the final diagnosis. In the first tier (stroke mimic vs. true stroke), true stroke patients were older and male-dominant. History of stroke risk factor were more frequent in the true stroke group, whereas history of seizure or psychiatric disease were less common. From clinical manifestations, "after awakening", lateralizing symptoms, "not ambulating", and "not able to grasp" were more prevalent in the true stroke group, while "mental change" was more frequent in the stroke mimic group. From vital-sign and initial laboratory findings, systolic and diastolic BP were higher in the true stroke group. In contrast, the stroke mimic group included more patients with low systolic BP (≤90mmHg) and extreme glucose level (initial glucose <80 or ≥400 mg/dl).

In the second tier (ischemic vs. hemorrhagic stroke), the ischemic stroke group was older and had a higher proportion of males than the hemorrhagic stroke group. The patients with hemorrhagic stroke had shorter onset-to-door time. History of stroke risk factor was more prevalent in the ischemic stroke group. "Sudden onset" was more frequent in the hemorrhagic stroke, while "after awakening" and "as unusual" were more common in the ischemic stroke. The ischemic stroke group showed more common "any asymmetry. "Mental change" was more prevalent in the hemorrhagic stroke group, while "abnormal sensation" was more frequent in the ischemic stroke group. Systolic and diastolic BP were higher in the hemorrhagic stroke group.

**Table 1. Clinical information, vital signs, and initial labs (CIVIL).**

| | Stroke- mimic (n = 406) | True stroke (n = 1,153) | p[a] | Ischemic stroke (n = 894) | Hemorrhagic stroke (n = 259) | p[b] |
|---|---|---|---|---|---|---|
| **Clinical information** | | | | | | |
| Age, years | 62.2 ± 15.8 | 64.5 ± 14.2 | 0.012 | 65.6 ± 14.0 | 60.5 ± 14.5 | <0.001 |
| Age ≥ 60 years, n (%) | 230 (56.7) | 743 (64.4) | 0.005 | 613 (68.6) | 130 (50.2) | <0.001 |
| Age ≤ 40 years, n (%) | 38 (9.4) | 57 (4.9) | 0.001 | 40 (4.5) | 17 (6.6) | 0.172 |
| Male, n (%) | 200 (49.3) | 699 (60.6) | <0.001 | 561 (62.8) | 138 (53.3) | 0.004 |
| Onset-to-door time (minute) | 172.6 ± 209.9 | 182.9 ± 205.2 | 0.118 | 197.9 ± 210.0 | 131.2 ± 178.7 | <0.001 |
| Onset-to-door ≥ 90 min, n (%) | 173 (42.6) | 466 (40.4) | 0.439 | 314 (35.1) | 152 (58.7) | <0.001 |
| Prior history, n (%) | | | | | | |
| Hypertension | 170 (41.9) | 606 (52.6) | <0.001 | 473 (52.9) | 133 (51.4) | 0.355 |
| Diabetes | 100 (24.6) | 250 (21.7) | 0.419 | 214 (23.9) | 36 (13.9) | <0.001 |
| Cardiac diseases | 69 (17.0) | 263 (22.8) | <0.001 | 244 (27.3) | 19 (7.3) | <0.001 |
| Previous stroke | 78 (19.2) | 242 (21.0) | 0.120 | 199 (22.3) | 43 (16.6) | 0.028 |
| Seizure or psychiatric history | 89 (21.9) | 21 (1.8) | <0.001 | 19 (2.1) | 2 (0.8) | 0.192 |
| Clinical manifestations | | | | | | |
| Time | | | | | | |
| "Sudden", n (%) | 365 (89.9) | 1038 (90.0) | 0.172 | 786 (87.9) | 252 (97.3) | <0.001 |
| "After awakening", n (%) | 32 (7.9) | 114 (9.9) | 0.021 | 105 (11.7) | 9 (3.5) | <0.001 |
| "As unusual", n (%) | 13 (3.2) | 34 (2.9) | 0.476 | 31 (3.5) | 3 (1.2) | 0.034 |
| Body-spatial | | | | | | |
| "one-side arm", n (%) | 144 (35.5) | 821 (71.2) | <0.001 | 642 (71.8) | 179 (69.1) | 0.221 |
| "one-side leg", n (%) | 120 (29.6) | 720 (62.4) | <0.001 | 557 (62.3) | 163 (62.9) | 0.457 |
| "one-side face", n (%) | 72 (17.7) | 401 (34.8) | <0.001 | 306 (34.2) | 95 (36.7) | 0.255 |
| "any asymmetry', n (%) | 205 (50.5) | 958 (83.1) | <0.001 | 763 (85.3) | 195 (75.3) | <0.001 |
| Symptoms | | | | | | |
| "not ambulating", n (%) | 79 (19.5) | 508 (44.1) | <0.001 | 388 (43.4) | 120 (46.3) | 0.222 |
| "not able to speak", n (%) | 185 (45.6) | 576 (50.0) | 0.077 | 447 (50.0) | 129 (49.8) | 0.506 |
| "not able to grasp", n (%) | 37 (9.1) | 206 (17.9) | <0.001 | 164 (18.3) | 42 (16.2) | 0.245 |
| "mental change" *, n (%) | 156 (38.4) | 193 (16.7) | <0.001 | 92 (10.3) | 101 (39.0) | <0.001 |
| "abnormal sensation", n (%) | 56 (13.8) | 162 (14.1) | 0.247 | 138 (15.4) | 24 (9.3) | 0.006 |
| "visual disturbance", n (%) | 4 (1.0) | 18 (1.6) | 0.165 | 17 (1.9) | 1 (0.4) | 0.062 |
| **Baseline vital signs** | | | | | | |
| SBP, mmHg | 137.7 ± 28.9 | 151.1 ± 28.6 | <0.001 | 147.7 ± 25.4 | 163.0 ± 34.9 | <0.001 |
| SBP ≥ 160mmHg, n (%) | 100 (24.6) | 473 (41.0) | <0.001 | 327 (36.6) | 146 (56.4) | <0.001 |
| SBP ≥ 140mmHg, n (%) | 202 (49.8) | 801 (69.5) | <0.001 | 598 (66.9) | 203 (78.4) | <0.001 |
| SBP ≤ 90mmHg, n (%) | 12 (3.0) | 3 (0.3) | <0.001 | 1 (0.1) | 2 (0.8) | 0.128 |
| DBP, mmHg | 81.9 ± 34.7 | 86.8 ± 16.4 | 0.018 | 85.6 ± 15.5 | 90.6 ± 18.7 | 0.005 |
| Pulse rate, bpm | 84.2 ± 19.1 | 82.9 ± 15.5 | 0.454 | 82.8 ± 15.9 | 83.2 ± 13.9 | 1.000 |
| Body temperature, ˚C | 36.5 ± 0.8 | 36.5 ± 0.5 | 1.000 | 36.5 ± 0.4 | 36.4 ± 0.6 | 0.173 |
| **Initial laboratory findings** | | | | | | |
| Glucose, mg/dl | 150.0 ± 101.6 | 146.8 ± 60.1 | 0.510 | 144.0 ± 60.5 | 156.4 ± 57.8 | 0.051 |
| Extreme glucose level†, n (%) | 24 (5.9) | 14 (1.2) | <0.001 | 11 (1.2) | 3 (1.2) | 1.000 |
| Oxygen saturation, % | 99.3 ± 3.2 | 99.5 ± 2.5 | 0.062 | 99.6 ± 2.2 | 99.2 ± 3.2 | 0.082 |

SBP means systolic blood pressure, and DBP indicates diastolic blood pressure. *"mental change" was defined when a decrease in the level of consciousness below drowsiness was observed on initial neurological examination. †initial blood glucose level ≤ 80 or ≥ 400 mg/dl, p[a] = Stroke mimic vs. true stroke, p[b] = Ischemic stroke vs. hemorrhagic stroke

**Table 2. Clinical information, vital signs, and initial labs (CIVIL) between ELVO and non-ELVO patients.**

| | ELVO stroke (n = 291) | Non-ELVO stroke (n = 603) | p |
|---|---|---|---|
| **Clinical information** | | | |
| Age, years | 68.1 ± 13.6 | 64.4 ± 14.0 | <0.001 |
| Male, n (%) | 171 (58.8) | 390 (64.7) | 0.087 |
| Onset-to-door time (minute) | 191.8 ± 208.8 | 200.9 ± 210.7 | 0.045 |
| Prior history, n (%) | | | |
| Hypertension | 160 (55.0) | 313 (51.9) | 0.388 |
| Diabetes | 59 (20.3) | 155 (25.7) | 0.075 |
| Cardiac problems | 115 (39.5) | 129 (21.4) | <0.001 |
| Previous stroke | 57 (19.6) | 142 (23.5) | 0.182 |
| Manifestation | | | |
| Time domain | | | |
| "Sudden", n (%) | 260 (89.3) | 526 (87.2) | 0.363 |
| "After awakening", n (%) | 30 (10.3) | 75 (12.4) | 0.354 |
| "As unusual", n (%) | 10 (3.4) | 21 (3.5) | 0.972 |
| Body-spatial domain | | | |
| "one-side arm", n (%) | 241 (82.8) | 401 (66.5) | <0.001 |
| "one-side leg", n (%) | 217 (74.6) | 340 (56.4) | <0.001 |
| "one-side face", n (%) | 112 (38.5) | 194 (32.2) | 0.062 |
| "any asymmetry', n (%) | 257 (88.3) | 506 (83.9) | 0.081 |
| Symptom domain | | | |
| "not ambulating", n (%) | 159 (54.6) | 229 (38.0) | <0.001 |
| "not able to speak", n (%) | 188 (64.6) | 259 (43.0) | <0.001 |
| "not able to grasp", n (%) | 53 (18.2) | 111 (18.4) | 0.944 |
| "mental change" *, n (%) | 59 (20.3) | 33 (5.5) | <0.001 |
| "abnormal sensation", n (%) | 14 (4.8) | 124 (20.6) | <0.001 |
| "visual disturbance", n (%) | 9 (3.1) | 8 (1.3) | 0.070 |
| "gaze deviation", n (%) | 197 (67.7) | 48 (8.0) | <0.001 |
| **Baseline vital signs** | | | |
| SBP, mmHg | 143.1 ± 26.4 | 149.9 ± 24.7 | <0.001 |
| DBP, mmHg | 83.5 ± 16.0 | 86.7 ± 15.1 | 0.003 |
| Pulse rate, bpm | 84.2 ± 18.1 | 82.1 ± 14.7 | 0.064 |
| Body temperature, ˚C | 36.4 ± 0.4 | 36.5 ± 0.4 | 0.004 |
| **Initial laboratory findings** | | | |
| Glucose, mg/dl | 142.4 ± 49.3 | 144.8 ± 65.2 | 0.571 |
| Oxygen saturation, % | 99.5 ± 1.7 | 99.7 ± 2.4 | 0.202 |
| **Stroke characteristics** | | | |
| NIHSS, median (IQR) | 16 (12–20) | 3 (1–6) | <0.001 |
| TOAST classification, n (%) | | | <0.001 |
| Large artery disease | 69 (23.7) | 105 (17.4) | |
| Cardioembolism | 157 (54.0) | 106 (17.6) | |
| Small artery disease | 0 (0.0) | 134 (22.2) | |
| Others | 65 (22.3) | 258 (42.8) | |
| Vessel occlusion, n (%) | | | |
| ICA | 90 (30.9) | - | |
| M1 | 102 (35.1) | - | |
| M2 | 7 (16.2) | - | |
| BA | 33 (11.3) | - | |

(*Continued*)

**Table 2.** (Continued)

|  | ELVO stroke (n = 291) | Non-ELVO stroke (n = 603) | p |
|---|---|---|---|
| Others | 19 (6.5) | - | |
| tPA use, n (%) | 155 (53.3) | 83 (13.8) | <0.001 |
| Endovascular treatment, n (%) | 150 (51.5) | 0 (0.0) | <0.001 |

\*"mental change" was defined when a decrease in the level of consciousness below drowsiness was observed on initial neurological examination. SBP = systolic blood pressure, DBP = diastolic blood pressure, NIHSS = National Institute of Health Stroke Scale, ICA = internal carotid artery, BA = basilar artery, tPA = tissue plasminogen activator

## Tier 1: Stroke mimic vs. true stroke: CIVIL-$AS^3A^2P$

We determined independently significant factors in the CIVIL system using multiple regression analysis (Fig 1A). _A_ge (≥60 years), _S_troke risks (history of cardiac disease), "any _A_symmetry", and "not _A_mbulating" were positive discriminatory items for diagnosis of true stroke, while younger _A_ge (≤40 years), history of _S_eizure or psychiatric disease were negative discriminatory items. In vital signs and laboratory data, high B_P_ (systolic BP ≥140mmHg) was a positive discriminatory item, and low B_P_ (systolic BP ≤90mmHg) and extreme _S_ugar level (≤80 or ≥400 mg/dL) were included as negative discriminatory items. Asymmetric leg weakness, non-lateralizing symptoms, mental change, and initial oxygen saturation were indiscriminate. The CIVIL-$AS^3A^2P$ score was finally determined as overall 7 items (Fig 2): 5 clinical items, 1 vital sign, and 1 initial laboratory finding. The total score ranged from -5 to +6. Retrospective validation on 1,559 suspected stroke patients determined an optimal cut-off point for stroke diagnosis as ≥ +1. At this cut-off point, the diagnostic performance of CIVIL-$AS^3A^2P$ score was as follows: sensitivity 82.1%, specificity 56.4%, positive predictive value (PPV) 84.3%, and negative predictive value (NPV) 52.6% (Youden's index 0.385). We compared the performance of the CIVIL-ASAP score to the CPSS, LAPSS, and ROSIER scales in our data set. The sensitivity and specificity of these established recognition systems were 90.4% and 29.1% in CPSS, 69.7% and 67.7% in LAPSS, 93.8% and 34.0% in ROSIER. In ROC curve analysis, CIVIL-$AS^3A^2P$ score had a superior diagnostic performance than the other three systems per area under the curve (S3 Fig, 0.767 in CIVIL-ASAP vs. 0.751 in ROSIER vs. 0.687 in LAPSS vs. 0.597 in CPSS). Comparisons among early stroke recognition scales were described in Fig 3.

## Tier 2: Ischemic vs. hemorrhagic stroke: CIVIL-_MAPS_

To differentiate between ischemic and hemorrhagic stroke, second tier analysis using multiple regression was conducted (Fig 1B). _MAPS_: _M_ental change, _A_ge below 60 years, high blood _P_ressure (systolic BP ≥160mmHg), no _S_troke risk (without history of diabetes or cardiac disease) were positive discriminatory items for diagnosis of hemorrhagic stroke (Fig 2). The CIVIL-_MAPS_ score consisted of 3 clinical items and 1 vital sign, and the total score ranged from 0 to +4. Retrospective validation on the 1,153 true stroke patients determined an optimal cut-off point for hemorrhagic stroke diagnosis as ≥ 2. At this cut-off point, the diagnostic performance of CIVIL-_MAPS_ score was as follows: sensitivity of 47.5%, specificity of 85.4%, PPV of 50.6%, NPV of 83.8% (Youden's index 0.329).

## Tier 3: Non-ELVO vs. ELVO: CIVIL-_GFAST_

In the final tier, we applied the GFAST score to select ELVO patients (Fig 1C). The score was calculated as the sum of positive symptoms: _G_aze deviation, _F_ace asymmetry, _A_rm asymmetry, and _S_peech disturbance (Fig 2). Retrospective validation on 894 ischemic stroke patients

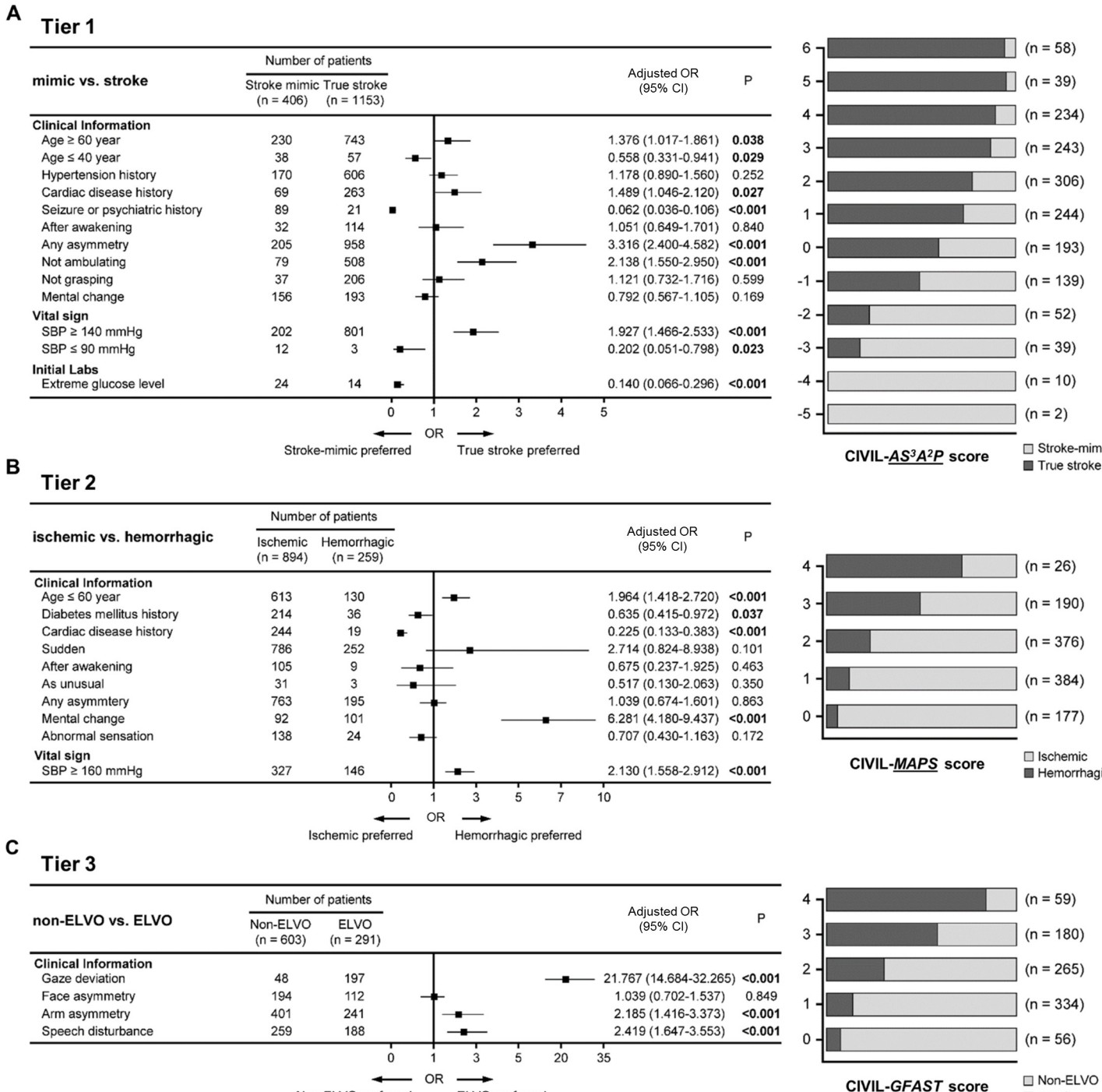

**Fig 1. Results of multiple regression analysis and distribution of patients according to the CIVIL scores.** (A) Tier 1: CIVIL-$AS^3A^2P$ score consisted of Age ($\leq$40 years or $\geq$60 years), Stroke risk (cardiac disease history) without Seizure or psychiatric history, extreme Sugar level ($\leq$80 or $\geq$400mg/dl), any Asymmetry, not Ambulating, and Pressure (SBP$\leq$90 mmHg or $\geq$140mmHg). (B) Tier 2: CIVIL-MAPS included Mental change, Age ($\leq$60 years), Pressure (SBP$\geq$160mmHg), and no Stroke risks (history of diabetes mellitus or cardiac disease). (C) Tier 3: To identify emergent large vessel occlusion patients, GFAST score was incorporated into the CIVIL scoring system (Gaze, Face asymmetry, Arm asymmetry, Speech disturbance).

| Assessment | CIVIL | CPSS | LAPSS | ROSIER |
|---|---|---|---|---|
| **Clinical Information** | | | | |
| Age | ✔ (≥60 or ≤40 years) | - | ✔ (>45 years) | - |
| Premorbid status | - | - | ✔ (mRS <4) | - |
| Hypertension | - | - | - | - |
| Cardiac disease | ✔ | - | - | - |
| Seizure history | ✔ | - | ✔ (no history of seizure) | ✔ (no seizure at onset, No LOC or syncope at onset) |
| Psychiatric history | ✔ | - | - | - |
| Elapsed time | ✔ (<6 hours) | - | ✔ (≤25 hours) | - |
| Facial droop | ✔ (any asymmetry) | ✔ | ✔ | ✔ |
| Arm weakness | ✔ (any asymmetry) | ✔ | ✔ | ✔ |
| Handgrip | - | - | ✔ | - |
| Leg weakness | ✔ (any asymmetry) | - | - | ✔ |
| Not able to walk | ✔ | - | - | - |
| Speech difficulty | - | ✔ | - | ✔ |
| Visual field | - | - | - | ✔ |
| **Vital signs** | | | | |
| Blood pressure | ✔ (SBP≥140 or ≤90mmHg) | - | - | - |
| **Initial Labs** | | | | |
| Blood glucose | ✔ (≤80 or ≥400mg/dl) | - | ✔ (2.8 to 22.2mmol/L) | ✔ (>3.5mmol/L) |

**Fig 2. Summary of items and scoring in the CIVIL system.** In tier 1 (CIVIL-$AS^3A^2P$), 7 items which included clinical information (white), vital signs (grey), and initial labs (dark grey) were used. Stroke-preferred items were assigned positive points and stroke mimic preferred items were negative points (ranged from −5 to +6). Tier 2 (CIVIL-*MAPS*) allocated 4 items with clinical information and vital signs, and the GFAST system was applied in tier 3 (CIVIL-*GFAST*) for the selection of ELVO patients.

determined the optimal cut-off point for ELVO diagnosis as ≥ 3. At this cut-off point, the diagnostic performance of CIVIL-*GFAST* score was as follows: sensitivity of 66.5%, specificity of 79.8%, PPV of 54.6%, NPV of 86.7% (Youden's index 0.463). The CIVIL scoring system is summarized in Fig 2.

## Discussion

Our data support that the CIVIL scoring system is feasible for identifying suspicious acute stroke patients in a stepwise fashion: true stroke or not, hemorrhagic stroke or not, and ELVO or not. In addition, step-by-step acronyms ($AS^3A^2P$, *MAPS*, and *GFAST*) can be used in a wide range of fields of prehospital and ER-based situations to serve as triaging tools for patients to be easily remembered.

The CIVIL scoring system can help us to differentiate different types of stroke at the same time. Acute stroke is an urgent condition that requires rapid evaluation and proper management because the longer a stroke goes untreated, the greater the brain damage (time is brain) [22]. Efficient triaging is important for acute stroke patients to guide proper disposition and early interventions, which may be entirely decisive in some cases [23,24]. Due to limited time window for thrombolytic therapy, numerous prehospital scoring systems for early recognition of ischemic stroke have been developed [8,14,15]. Recently, endovascular recanalization therapy in ELVO patients has been proven as the standard treatment, consequently, several scoring

| Tier 1: mimic vs stroke | Mimic preferred | Items (CIVIL-*AS³A²P*) | Stroke preferred |
|---|---|---|---|
| **C**linical Information | ≤ 40 years (-1) | **A**ge | ≥ 60 years (+1) |
| | No (-1) | **S**troke risks (cardiac) | Yes (+1) |
| | Yes (-1) | **S**eizure or psychiatric history | No (+1) |
| **V**ital signs | ≤ 80 or ≥ 400mg/dl (-1) | **S**ugar | - |
| | No (-1) | **A**symmetry | Yes (+1) |
| Initial **L**abs | No (-1) | not **A**mbulating | Yes (+1) |
| | ≤ 90 mmHg (-1) | **P**ressure (SBP) | ≥140 mmHg (+1) |
| **Tier 2: ischemic vs. hemorrhagic** | **Ischemic preferred** | **Items (CIVIL-*MAPS*)** | **Hemorrhagic preferred** |
| **C**linical Information | No | **M**ental change | Yes (+1) |
| | No | **A**ge < 60 years | Yes (+1) |
| **V**ital signs | No | **P**ressure (SBP ≥ 160mmHg) | Yes (+1) |
| | Yes | **S**troke risks (DM, cardiac) | No (+1) |
| **Tier 3: non-ELVO vs. ELVO** | **Non-ELVO preferred** | **Items (CIVIL-*GFAST*)** | **ELVO preferred** |
| **C**linical Information | No | **G**aze deviation | Yes (+1) |
| | No | **F**ace asymmetry | Yes (+1) |
| | No | **A**rm asymmetry | Yes (+1) |
| | No | **S**peech disturbance | Yes (+1) |

**Fig 3. Descriptive comparison of various early stroke recognition scales.** CIVIL = Clinical Information, Vital signs, and Initial Labs, CPSS = Cincinnati Prehospital Stroke Scale, LAPSS = Los Angeles Prehospital Stroke Screen, ROSIER = Recognition Of Stroke In the Emergency Room.

systems to recognize ELVO have been addressed [19–21]. However, current scoring systems have focused only on one aspect of stroke identification: stroke versus stroke mimic, or ELVO discrimination [25]. In addition, little has been elucidated to distinguish hemorrhagic stroke from ischemic stroke especially in situations with limited imaging facilities [26]. To the best of our knowledge, there has been no definitive scoring system that integrates various aspects of stroke diagnosis.

This scoring system features a stepwise approach to triage stroke suspicious patients. When paramedics in emergency medical service (EMS) or ER physicians proceed step by step, they will be able to properly classify stroke patients who require rapid treatment. CIVIL-*AS³A²P* initially differentiates patients with true stroke from stroke mimics. In this first tier, it is important not to exclude potential patients who need. In this context, as we expected, CIVIL-*AS³A²P* showed relatively high sensitivity and low specificity for including all possible candidates. Second (CIVIL-*MAPS*) and third tiers (CIVIL-*GFAST*) showed low sensitivity and high specificity so that patients in need of urgent treatments (thrombolysis and/or endovascular therapy) could be selected effectively. The CIVIL scoring system enables rapid identification of patients delivered to the ER with high sensitivity to identify the actual stroke, and also enables the recognition of hemorrhagic stroke and ELVO with high specificity.

The CIVIL scoring system included objective vital signs and laboratory findings as well as clinical information. Previous scoring systems consisting of clinical manifestations may be affected by the examiner's experience and special training is needed to reduce inter-observer variability [27]. Some validation studies on early recognition scoring systems reported high variability in inter-observer reliability ranging from 69% to 90% [14,28]. To compensate for these variations, ROSIER [8] and LAPSS [15] included laboratory finding such as blood glucose levels. For this reason, the CIVIL scoring system contained vital signs in addition to laboratory findings designed to apply more objective parameters. Moreover, the extreme values of vital signs and laboratory findings-blood glucose level $\leq$ 80 or $\geq$ 400mg/dl and systolic BP $\geq$140 mmHg or $\leq$ 90 mmHg help to discriminate stroke mimic conditions such as sepsis, shock, or syncope. Therefore, our new scoring system could overcome some potential limitations of other previous scoring systems by including objective and quantitative items.

In this study, the CIVIL scoring system was developed for use in both ER and prehospital settings. The selection of acute stroke patients in the prehospital and emergent setting continue to be the subject of research due to the time-dependent nature of stroke [27]. Various early recognition systems have been used in the EMS to properly transport stroke patients to more appropriate centers. However, there have been several limitations including inconvenience, imperfect accuracy, and time-consuming training [19,20]. Moreover, an increase in items adds complexity to the system for rapid evaluation [29]. The CIVIL scoring system applies an intuitive and easy-to-remember acronym for EMS and other medical professionals to be easily used. We applied simple and familiar GFAST to improve accessibility in the third tier for the identification of ELVO patients.

Our study has some limitations. First, it is an observational study with retrospective nature; however, all information has been automatically stored in a prospectively-collecting database at a large regional emergency and stroke center. Second, data with time windows over 6 hours were not covered in the current study. From recent trials, mechanical thrombectomy is indicated up to 24 hours after stroke onset. Nevertheless, most patients with onset to treatment time less than 6 hours need more urgent treatment regardless of core-penumbra mismatch, so that our recognition system can more properly apply to those patients. Third, there are limits in the conclusions that can be drawn regarding the performance of the CIVIL system in patients with posterior circulation acute ischemic stroke. This scoring system was designed to focus on patients with anterior circulation which is supported by the current guideline for endovascular treatment. Finally, the sensitivity of tier 2 and 3 are less than 80%, the results should be interpreted with caution. In the future, prospective validation of the CIVIL scoring system should include a systematic education program for paramedics to improve performance.

In conclusion, the CIVIL scoring system can be used as a comprehensive and versatile tool to recognize true stroke and identify stroke subtypes simultaneously.

## Supporting information

**S1 STROBE Checklist.**
(DOC)

**S1 Data. Deidentified raw dataset.**
(XLSX)

**S1 Fig. The Korean version of EMR based matrix for stroke suspicious patients.** (Cubic S model)
(PDF)

**S2 Fig. Flow diagram of 1,621 suspicious stroke patients.**
(PDF)

**S3 Fig. Receiver-Operating Characteristic (ROC) curve and corresponding area under the curve (AUC) statistics of the CIVIL scoring system.**
(PDF)

## Author Contributions

**Conceptualization:** Ji Man Hong.

**Data curation:** Sung Eun Lee, Hyo Jung Kang.

**Formal analysis:** Sung Eun Lee, Mun Hee Choi.

**Investigation:** Sung Eun Lee, Hyo Jung Kang, Seong-Joon Lee.

**Methodology:** Sung Eun Lee, Mun Hee Choi, Ji Man Hong.

**Project administration:** Ji Man Hong.

**Resources:** Seong-Joon Lee, Jin Soo Lee.

**Supervision:** Jin Soo Lee, Yunhwan Lee, Ji Man Hong.

**Validation:** Mun Hee Choi, Seong-Joon Lee.

**Visualization:** Mun Hee Choi.

**Writing – original draft:** Sung Eun Lee.

**Writing – review & editing:** Mun Hee Choi, Jin Soo Lee, Yunhwan Lee, Ji Man Hong.

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
