## [Decision Letter · Decision Letter 0]

20 Jan 2020

PONE-D-19-32649

Stepwise stroke recognition through Clinical Information, Vital signs, and Initial Labs (CIVIL): Electronic health record-based observational cohort study

PLOS ONE

Dear Dr. Hong,

Thank you for submitting your manuscript to PLOS ONE. After careful consideration, we feel that it has merit but does not fully meet PLOS ONE’s publication criteria as it currently stands. Therefore, we invite you to submit a revised version of the manuscript that addresses the points raised during the review process.

We would appreciate receiving your revised manuscript by Mar 05 2020 11:59PM. To enhance the reproducibility of your results, we recommend that if applicable you deposit your laboratory protocols in protocols.io, where a protocol can be assigned its own identifier (DOI) such that it can be cited independently in the future. For instructions see: http://journals.plos.org/plosone/s/submission-guidelines#loc-laboratory-protocols

We look forward to receiving your revised manuscript.

Kind regards,

Juan Manuel Marquez-Romero, M.D., M.Sc.

Academic Editor

PLOS ONE

Reviewers' comments:

1. Is the manuscript technically sound, and do the data support the conclusions?

Reviewer #1: Yes

Reviewer #2: Yes

2. Has the statistical analysis been performed appropriately and rigorously? 

Reviewer #1: No

Reviewer #2: Yes

3. Have the authors made all data underlying the findings in their manuscript fully available?

Reviewer #1: Yes

Reviewer #2: Yes

4. Is the manuscript presented in an intelligible fashion and written in standard English?

Reviewer #1: No

Reviewer #2: Yes

5. Review Comments to the Author

Reviewer #1: This paper is interesting, but the sensitivity results are less than 80% for tier 2 and 3. The stroke is a serious disease with potential treatment therefore requires better levels of sensitivity. I recommend mentioning this finding as a weakness of the study.

Reviewer #2: I had the pleasure to read and review the manuscript: "Stepwise stroke recognition through Clinical Information, Vital signs, and Initial Labs (CIVIL): Electronic health record-based observational cohort study", which is a very interesting article addressing the main objective of develop a scoring system for three different scenarios: ischemic stroke, hemorrhagic stroke and LVO, in a pre-hospital stage. Before considering this manuscript for publication, I have some comments to add:

1) Abstract:

Nothing to add

2) Introduction:

One of the main aspects that should be mentioned in this manuscript, is the current availability of scales and scoring systems in the pre-hospital management of possible stroke cases. Two recent articles (Stroke. 2019;50:e285–e286 and Stroke. 2015 Jun;46(6):1513-7) already intended to analyze the complexity of the different systems that are currently part of many emergency and pre-hospital departments around the world. The justification of doing your research, seems that optimal and more valid scoring systems are needed, but I can't see this rationale the way you are presenting the manuscript.

2) Methods:

- Data availability (repository or by request) should be clarified.

- I understand that information to develop this scoring system is based on information from the Emergency Department, mainly from data that already is being recruited from the Cubic S model, but all this information is captured AT the emergency department, and was transferred to the pre-hospital scenario, which seems logic, but also is a different area of work and the possibility of losing information from the real pre-hospital work is plausible.

- Statistical analysis: which variables were included at the multivariate analysis? Were they pre-defined? If so, which cut-off point was decided to include the variable in the multivariate model.

- Youden's Index together with the scoring system performance should be included.

- OR and CI 95% are included in your analysis, but this also should be mentioned at the "statistics" section

- As you are doing only internal validation of your scoring system, did you consider a bootstraping to evaluate a more accurate performance in your population?

3) Results

- The decision on each variable included in the CIVIL ASAP tool, was done based on current, recent and the highest level of bibliography, or only extracted from the dataset.

- How do you define "mental change"? In terms of the MAPS scoring system?

- I can't see any of the results referring to the multivariate model... Do all the OR are un-adjusted or adjusted? If you adjusted, which variables were included at the model?

- Did you perform a ROC-AUC analysis to evaluate the performance of your scoring system compared to the other systems (you mention that at the methods section)? Could you provide a figure of the comparison of each curve according to the pre-hospital score used to recognize a "confirmed stroke case"

4) Discussion:

- Only one comment: current scoring systems are very easy to use; its performance vary, and seems that the CIVIL has a very good opportunity to prove your rationale, but I think that the applicability of the scoring point is very difficult, so, you should try to convince the readers that they should use this system.

---

## [Author Response · Author response to Decision Letter 0]

19 Feb 2020

Reviewer #1: 

Comment 1. This paper is interesting, but the sensitivity results are less than 80% for tier 2 and 3. The stroke is a serious disease with potential treatment therefore requires better levels of sensitivity. I recommend mentioning this finding as a weakness of the study.

Response 1. Thank you for the reviewer’s valuable comment. We totally agree with the reviewer’s comment. As the reviewers mentioned, the sensitivity of CIVIL score is limited, and the results should be interpreted with caution. The CIVIL scoring system had a relative high sensitivity in tier 1 (to identify all possible candidates), and a relative high specificity in tier 2 and 3 (to recognize the need for urgent treatments, such as intravenous thrombolysis or endovascular thrombectomy). Therefore, we added limitation of the sensitivity results in discussion section of the revised manuscript (p20-1). Please check this point.

 

Reviewer #2: I had the pleasure to read and review the manuscript: "Stepwise stroke recognition through Clinical Information, Vital signs, and Initial Labs (CIVIL): Electronic health record-based observational cohort study", which is a very interesting article addressing the main objective of develop a scoring system for three different scenarios: ischemic stroke, hemorrhagic stroke and LVO, in a pre-hospital stage. Before considering this manuscript for publication, I have some comments to add:

1) Abstract:

Nothing to add

2) Introduction: 

Comment 1. One of the main aspects that should be mentioned in this manuscript, is the current availability of scales and scoring systems in the pre-hospital management of possible stroke cases. Two recent articles (Stroke. 2019;50:e285–e286 and Stroke. 2015 Jun;46(6):1513-7) already intended to analyze the complexity of the different systems that are currently part of many emergency and pre-hospital departments around the world. The justification of doing your research, seems that optimal and more valid scoring systems are needed, but I can't see this rationale the way you are presenting the manuscript.

Response 1. Thank you for the reviewer’s valuable comment. Various scales and scoring systems have been developed and analyzed, but there is still no consensus on which scale work better, especially in large vessel occlusion (LVO).1,2 In addition, the accuracy of EMS stroke recognition is not enough due to issues related to false positives and false negatives.3 As noted by the reviewer, the need to develop a more optimized and valid scoring system should have been more convincing. Therefore, we added this rationale in the introduction section of the revised manuscript (p5). Please consider this point.

1. Walker GB, Zhelev Z, Henschke N, et al. Prehospital Stroke Scales as Screening Tools for Early Identification of Stroke and Transient Ischemic Attack. Stroke. 2019;50(10):e285-e286.

2. Zhelev Z, Walker G, Henschke N, et al. Prehospital stroke scales as screening tools for early identification of stroke and transient ischemic attack. Cochrane Database Syst Rev. 2019;4:CD011427.

3. Oostema JA, Konen J, Chassee T, et al. Clinical predictors of accurate prehospital stroke recognition. Stroke. 2015;46(6):1513-7.

2) Methods:

Comment 2. Data availability (repository or by request) should be clarified.

Response 2. In accordance with journal requirements, we added an unidentified data set as supporting information. Please check this point. 

Comment 3. I understand that information to develop this scoring system is based on information from the Emergency Department, mainly from data that already is being recruited from the Cubic S model, but all this information is captured at the emergency department, and was transferred to the pre-hospital scenario, which seems logic, but also is a different area of work and the possibility of losing information from the real pre-hospital work is plausible.

Response 3. We fully understand the reviewer’s concern about the applicability of the CIVIL scoring system to real pre-hospital work. As the reviewer mentioned, our data were collected in the emergency department by ER physicians. In detail, the items consist of clinical information, vital signs, and initial labs. The selected items were age, past history, key clinical manifestations, vital sign, and blood glucose level. In actual prehospital situations, clinical manifestations (asymmetry, not ambulating, gaze deviation, speech disturbance) are familiar and have often been used in other well-known scoring systems (CPSS, LAPSS, ROSIER, etc.). Also, the vital signs and blood sugar levels are always checked in ambulances. Therefore, we believe that the CIVIL scoring system can be easily applied to actual pre-hospital work after systematically educating EMS paramedics. In the discussion section of the revised manuscript (p21), we described that EMS paramedics will require systematic training and verification. Please check this point.

Comment 4. Statistical analysis: which variables were included at the multivariate analysis? Were they pre-defined? If so, which cut-off point was decided to include the variable in the multivariate model.

Response 4. We entered variables with p value < 0.05 in univariate analyses into a multivariate logistic regression model. For confounding factors, more significant or intuitive variables were selected. Please check this point in the revised manuscript (p10).

Comment 5. Youden's Index together with the scoring system performance should be included.

Response 5. Thank you for the reviewer’s comment. Youden’s index was 0.385 at a cut-off point ≥ 1 in tier 1, 0.329 at a cut-off point ≥ 2 in tier 2, and 0.463 at a cut-off point ≥ 3 in tier 3, respectively. We have provided these values in the results section of the revised manuscript (p16-7). Please check this point.

Comment 6. OR and CI 95% are included in your analysis, but this also should be mentioned at the "statistics" section

Response 6. We added OR and CI 95% analysis in the statistics section of the revised manuscript (p10). Please check this point. 

Comment 7. As you are doing only internal validation of your scoring system, did you consider a bootstraping to evaluate a more accurate performance in your population?

Response 7. Thank you for the reviewer’s valuable comment. As the reviewer’s comment, we performed additional analysis using bootstrap to evaluate more accurate performance. Bootstrap results showed that the parameter estimates closely agreed with the corresponding values in the final model (Theses results were based on 1,000 bootstrap sample.). Please check the tables below.

Tier 1

 Model estimates Bootstrap results

Variables β (SE%) OR with 95% CI P β (SE%) OR with 95% CI P

Age ≥ 60 years 0.32 (15.4) 1.38 (1.02-1.86) 0.04 0.32 (15.1) 1.38 (1.03-0.87) 0.03

Age ≤ 40 years -0.58 (26.6) 0.56 (0.33-0.94) 0.03 -0.58 (28.8) 0.56 (0.32-1.00) 0.04

Hypertension history 0.16 (14.3) 1.18 (0.89-1.56) 0.25 0.16 (14.2) 1.18 (0.89--1.54) 0.24

Cardiac disease history 0.40 (18.0) 1.49 (1.05-2.12) 0.03 0.40 (18.2) 1.49 (1.05-2.13) 0.03

Seizure or psychiatric history -2.79 (27.8) 0.06 (0.04-0.11) <0.01 -2.79 (29.3) 0.06 (0.03-0.10) 0.00

After awakening 0.05 (24.6) 1.05 (0.65-1.70) 0.84 0.05 (25.5) 1.05 (0.63-1.76) 0.84

Any asymmetry 1.20 (16.5) 3.32 (2.40-4.58) <0.01 1.20 (16.9) 3.32 (2.40-4.68) 0.00

Not ambulating 0.76 (16.4) 2.14 (1.55-2.95) <0.01 0.76 (16.6) 2.14 (1.54-2.98) 0.00

No grasping 0.11 (21.7) 1.12 (0.73-1.72) 0.60 0.11 (23.0) 1.12 (0.74-1.83) 0.62

Mental change -0.23 (17.0) 0.79 (0.57-1.11) 0.17 -0.23 (16.8) 0.79 (0.57-1.12) 0.16

SBP ≥ 140 mmgHg 0.66 (14.0) 1.93 (1.47-2.53) <0.01 0.66 (14.6) 1.93 (1.46-2.55) 0.00

SBP ≤ 90 mmHg -1.60 (70.2) 0.20 (0.05-0.80) 0.02 -1.60 (455.3) 0.20 (0.00-0.84) 0.03

Extreme glucose level -1.97 (38.3) 0.14 (0.07-0.30) <0.01 -1.97 (42.7) 0.14 (0.06-0.30) 0.00

Tier 2

 Model estimates Bootstrap results

Variables β (SE%) OR with 95% CI P β (SE%) OR with 95% CI P

Age ≤ 60 years 0.68 (16.6) 1.96 (1.42-2.72) <0.01 0.68 (16.9) 1.96 (1.43-2.77) 0.00

Diabetes mellitus history -0.45 (21.7) 0.64 (0.42-0.97) 0.04 -0.45 (22.0) 0.64 (0.41-0.95) 0.03

Cardiac disease history -1.49 (27.0) 0.23 (0.13-0.38) <0.01 -1.49 (27.6) 0.23 (0.11-0.36) 0.00

Sudden 1.00 (60.8) 2.71 (0.82-8.94) 0.10 1.00 (65.6) 2.71 (0.80-9.84) 0.08

After awakening -0.39 (53.5) 0.68 (0.24-1.93) 0.46 -0.39 (57.0) 0.68 (0.18-1.71) 0.45

As unusual -0.66 (70.6) 0.52 (0.13-2.06) 0.35 -0.66 (436.7) 0.52 (0.00-2.30) 0.40

Any asymmetry 0.04 (22.1) 1.04 (0.67-1.60) 0.86 0.04 (22.3) 1.04 (0.69-1.68) 0.85

Mental change 1.84 (20.8) 6.28 (4.18-9.44) <0.01 1.84 (22.1) 6.28 (4.25-10.08) 0.00

Abnormal sensation -0.35 (25.4) 0.71 (0.43-1.16) 0.17 -0.35 (27.2) 0.71 (0.39-1.14) 0.20

SBP ≥ 160 mmgHg 0.76 (15.9) 2.13 (1.56-2.91) <0.01 0.76 (16.6) 2.13 (1.56-3.00) 0.00

Tier 3

 Model estimates Bootstrap results

Variables β (SE%) OR with 95% CI P β (SE%) OR with 95% CI P

Gaze deviation 3.08 (20.1) 21.77 (14.68-32.27) <0.01 3.08 (21.4) 21.76 (15.04-34.99) 0.00

Face asymmetry 0.04 (20.0) 1.04 (0.70-1.54) 0.85 0.04 (20.3) 1.04 (0.69-1.55) 0.85

Arm asymmetry 0.78 (22.1) 2.19 (1.42-3.37) <0.01 0.78 (22.2) 2.19 (1.47-3.43) 0.00

Speech disturbance 0.88 (19.6) 2.421 (1.65-3.55) <0.01 0.88 (19.5) 2.421 (1.66-3.59) 0.00

3) Results:

Comment 8. The decision on each variable included in the CIVIL ASAP tool, was done based on current, recent and the highest level of bibliography, or only extracted from the dataset.

Response 8. The decision to include variables is important and critical in developing a scoring system. Various aspects had to be considered, such as previous systems, statistical results, and availability in prehospital and emergency room. In tier 1 (CIVIL-ASAP tool), multivariate analysis indicated that nine variables are independent prognostic indicators: age (≥ 60 years, ≤ 40 years), cardiac disease history, seizure or psychiatric history, any asymmetry, not ambulating, systolic BP (≥ 140 mmHg, ≤ 90 mmHg), and extreme glucose level (≤ 80 or ≥ 400 mg/dl). In tier 2 (CIVIL-MAPS), five parameters were independent variables (age ≤ 60 years, diabetes mellitus, cardiac disease, mental change, SBP ≥ 160 mmHg). We developed CIVIL scoring system based on statistical results. These variables are also well-recognized because they are frequently used in previous scoring systems. Please consider this point.

Comment 9. How do you define "mental change"? In terms of the MAPS scoring system?

Response 9. As the reviewer mentioned, the definition of “mental change” needs to be clarified. After the training of ER physicians, a decrease in the level of consciousness below drowsiness during the initial neurological examination was assigned to “mental change”. It is equivalent to a NIHSS level of consciousness score (1a) ≥ 1. We clarified the definition of mental change in the Table 1 and 2 of the revised manuscript (p14-5). Please check this point.

Comment 10. I can't see any of the results referring to the multivariate model... Do all the OR are un-adjusted or adjusted? If you adjusted, which variables were included at the model?

Response 10. Thank you for the reviewer’s comment. Due to spatial limitations, the results of the multivariate model are shown only in Figure 1. Multivariate logistic regression analyze was performed after adjusting significant variables in univariate analyses (P<0.05). We clarified this point in Figure 1 (adjusted OR). Please check this point.

Comment 11. Did you perform a ROC-AUC analysis to evaluate the performance of your scoring system compared to the other systems (you mention that at the methods section)? Could you provide a figure of the comparison of each curve according to the pre-hospital score used to recognize a "confirmed stroke case"

Response 11. The results of ROC-AUC analysis were shown in the supplementary figure (S3) of supporting information. In Tier 1, CIVIL-ASAP score was compared with CPSS, LAPSS, and ROSIER scores. Please consider this point.

4) Discussion:

Comment 12. Only one comment: current scoring systems are very easy to use; its performance vary, and seems that the CIVIL has a very good opportunity to prove your rationale, but I think that the applicability of the scoring point is very difficult, so, you should try to convince the readers that they should use this system. 

Response 12. As the reviewer have mentioned, the CIVIL scoring system appears to be more complex than current scoring systems. Usability is also important, but more detailed items may be required to improve the accuracy of the scoring system. Although the CIVIL scoring system included a significant number of items and had a stepwise approach, included items were frequently used in other well-known scoring systems (Figure 2). We also used an intuitive and easy-to-remember acronyms, and existing GFAST score was applied to Tier 3. These efforts can improve the accessibility of the scoring system. The complexity of the CIVIL scoring system can be overcome with systematic education and practical tools (such as mobile applications, checklist and automatic calculation system). We added need for further investigation to increase applicability of the CIVIL scoring system in the discussion section of the revised manuscript (p21). Please consider this point.

---

## [Decision Letter · Decision Letter 1]

17 Mar 2020

Stepwise stroke recognition through Clinical Information, Vital signs, and Initial Labs (CIVIL): Electronic health record-based observational cohort study

PONE-D-19-32649R1

Dear Dr. Hong,

We are pleased to inform you that your manuscript has been judged scientifically suitable for publication and will be formally accepted for publication once it complies with all outstanding technical requirements.

With kind regards,

Juan Manuel Marquez-Romero, M.D., M.Sc.

Academic Editor

PLOS ONE

Additional Editor Comments (optional):

Reviewers' comments:

Reviewer's Responses to Questions

**Comments to the Author**

1. If the authors have adequately addressed your comments raised in a previous round of review and you feel that this manuscript is now acceptable for publication, you may indicate that here to bypass the “Comments to the Author” section, enter your conflict of interest statement in the “Confidential to Editor” section, and submit your "Accept" recommendation.

Reviewer #2: All comments have been addressed

2. Is the manuscript technically sound, and do the data support the conclusions?

Reviewer #2: Yes

3. Has the statistical analysis been performed appropriately and rigorously? 

Reviewer #2: Yes

4. Have the authors made all data underlying the findings in their manuscript fully available?

Reviewer #2: Yes

5. Is the manuscript presented in an intelligible fashion and written in standard English?

Reviewer #2: Yes

6. Review Comments to the Author

Reviewer #2: No comments, all the suggestions were addressed therefore I consider it is suitable for publication

7. PLOS authors have the option to publish the peer review history of their article (what does this mean?). If published, this will include your full peer review and any attached files.

Reviewer #2: Yes: Miguel A. Barboza

---

## [Editor Report · Acceptance letter]

25 Mar 2020

PONE-D-19-32649R1 

Stepwise stroke recognition through Clinical Information, Vital signs, and Initial Labs (CIVIL): Electronic health record-based observational cohort study 

Dear Dr. Hong:

I am pleased to inform you that your manuscript has been deemed suitable for publication in PLOS ONE. Congratulations! Your manuscript is now with our production department. 

With kind regards,

on behalf of

Dr. Juan Manuel Marquez-Romero 

Academic Editor

PLOS ONE